# Evaluation of Two Commercial ELISA Kits for Measuring Equine Serum Gastrin Compared to Radioimmunoassay

**DOI:** 10.3390/ani14202937

**Published:** 2024-10-11

**Authors:** Jessica R. Vokes, Kristene R. Gedye, Amy L. Lovett, Max C. de Kantzow, Ran Shan, Catherine M. Steel, Benjamin W. Sykes

**Affiliations:** 1School of Veterinary Science, Massey University, Palmerston North 4410, New Zealand; 2Department of Agriculture, Fisheries and Forestry, Canberra 2601, Australia; 3Department of Veterinary Clinical Services, The Hong Kong Jockey Club, Conghua Racecourse, Guangzhou 510900, China

**Keywords:** EGUS, gastric acid, hormones

## Abstract

**Simple Summary:**

Gastric disease in horses is common and has significant effects on welfare and performance. Gastrin is a hormone involved in the control of gastric acid secretion. Investigations into the effect of gastrin on gastric disease are important for further understanding of the disease and its treatment. However, there has only been one assay validated for the measurement of gastrin in horses. This study aimed to compare the performance of two commercially available, non-validated assays to the previously validated assay. The results showed poor performance of both non-validated assays, and do not support their use for gastrin measurement in future research.

**Abstract:**

Gastrin is an important hormone involved in gastric acid secretion. Despite its importance, validated methods other than radioimmunoassay (RIA) to assess serum gastrin concentrations in horses are lacking. This study aims to determine the agreement between ELISA and RIA in quantifying equine serum gastrin concentrations. Serum gastrin concentrations were quantified using two ELISA kits and RIA. Samples (196) from 14 horses at different time points were analyzed using one ELISA kit and RIA, selected samples (7) were analyzed using a second ELISA kit, and the correlation between methods was calculated. The level of agreement was analyzed by Bland-Altman analysis and differences between ELISA and RIA were plotted against averages for each sample. The Pearson correlation between gastrin concentrations measured by ELISA and the RIA was 0.27 and −0.32 for ELISA kit 1 and kit 2, respectively. Mean bias (ELISA-RIA) was 198.40 pg/mL (95% CI: −142.95–539.76) and −17.90 pg/mL (95% CI: −89.98–54.19) for ELISA kit 1 and kit 2, respectively. Measurements of horse gastrin by both ELISA methods were highly variable, with an unacceptable correlation to the reference method, RIA. Using non-validated ELISA methods to quantify horse gastrin cannot be recommended.

## 1. Introduction

Gastric mucosal diseases in horses are clinically significant due to the effect on welfare and performance [1,2]. Intra-gastric pH is an important factor in the development and healing of equine gastric mucosal diseases [1,2,3]. Gastric acid is produced by gastric parietal cells in the form of hydrochloric acid secreted by luminal proton pumps [4]. Gastric parietal cells are stimulated to produce acid predominantly by histamine, which is produced by enterochromaffin-like (ECL) cells under stimulation by gastrin, and directly by gastrin [4]. Gastrin is a peptide hormone produced in gastric G-cells in response to luminal proteins [4]. Gastrin is then released into the circulation and has a stimulatory and trophic effect on ECL cells [4], in addition to the effect on acid production [5]. These effects are regulated by negative feedback via somatostatin which is produced and released by gastric D-cells in the presence of low intra-gastric pH [5]. Somatostatin has a direct effect on gastric G-cells, reducing gastrin production, as well as on parietal cells, reducing intra-gastric acid secretion [5].

Proton pump inhibitors (PPIs) block parietal cell acid secretion, causing an increase in intra-gastric pH. This class of drugs has been widely used for decades in horses for the treatment and prevention of gastric disease [3]. With an increase in luminal pH during treatment with PPIs, there is a loss of somatostatin’s negative feedback on gastrin, leading to hypergastrinemia [5,6,7,8,9]. Concerns for the adverse effects of PPIs in horses, including rebound gastric hyperacidity (RGH) following PPI discontinuation driven by hypergastrinemia [10], have been raised. This has led to several investigations into equine gastrin concentrations in recent years [11,12,13,14,15,16].

The use of radioimmunoassay (RIA) for the quantification of equine serum gastrin has been previously validated [17]. However, increasing safety concerns of radiation exposure and regulatory restrictions have seen a global reduction in laboratories performing RIA [18]. This has limited the availability of RIA for analyzing veterinary samples, including equine gastrin. Likely a sequela of RIA’s reduced availability, ELISA-based techniques dominate current literature [12,14,16], despite the lack of independently published validation and without a discussion on the limitations of these assays.

Upon measuring equine serum gastrin as part of another study [11], the investigators noticed poor agreement between the ELISA and RIA used. This led to the investigation of other commercially available methods of equine gastrin determination.

The aim of this study was to compare measured serum gastrin concentrations determined by two commercially available, non-validated ELISA kits to the RIA method, using samples collected for other studies investigating hypergastrinemia in horses [11,19]. These authors hypothesized that measurements determined by the ELISA kits would be comparable to those determined by the reference standard, RIA.

## 2. Materials and Methods

### 2.1. Animals

Fourteen Thoroughbred geldings, between 6 and 12 (median 7.5) years of age, were sourced from the Hong Kong Jockey Club’s Conghua Racecourse Jockey Training School. Horses were included if considered normal on clinical and lameness examination and considered to be suitable for a conditioning program by the trainer. The horses were in simulated race training as part of two different studies [11,19], during which the horses were treated with omeprazole 2.28 g by mouth once daily for a total of 57 days over a 13-week period. The number of horses selected was based on the primary outcome for another study, as described elsewhere [19]. This study was approved by the Ethics Committee of the Hong Kong Jockey Club (protocol ERC/031/2021).

### 2.2. Sample Collection and Storage

Baseline blood samples were collected on day 0, before omeprazole treatment. After commencement of omeprazole treatment, blood samples were taken on days 7, 14, 21, 28, 35, 42, 49, and 56 of the treatment period. After discontinuing omeprazole treatment, blood samples were taken for an additional 5 weeks on days 63, 70, 77, 84, and 91. All blood samples were taken 5 minutes before the morning feeding; 10 mL of blood were collected by jugular venipuncture directly into a plain serum tube. The samples were separated, the serum was frozen within 4 hours and stored at −20°C at the Hong Kong Jockey Club. The samples were transported frozen to Massey University (New Zealand) where they were stored at −20°C until analysis. This project was started during the COVID-19 pandemic, with government and supply chain-imposed restrictions causing unexpected delays in sample gastrin determination by all methods. These delays are quantified below. Although human gastrin is stable in serum at −20 °C for up to 5 years [20], the stability of gastrin in equine serum has not been studied, to the authors’ knowledge.

### 2.3. ELISA Kit 1

Equine serum gastrin concentrations for all samples were analyzed using a commercially available ELISA kit (Horse Gastrin (GT) ELISA kit, MyBioSource Inc, San Diego, CA, USA). This kit utilizes a two-part sandwich ELISA method using rabbit-origin polyclonal capture and detection antibodies with recombinant partial length immunogens from E.coli. The partial length sequence is not disclosed by the manufacturers. The manufacturer reported an assay detection range of 31.2–1000 pg/mL with a sensitivity of 5.0 pg/mL, intra- and inter-assay CV <15% [21]. The analyses were performed by the primary investigator in a controlled laboratory with strict adherence to the manufacturer’s instructions.

All samples were measured over 3 plates that were processed and analyzed at three separate time points measuring optical densities. Gastrin concentrations were calculated using individual standard curves according to the manufacturer’s instructions. Briefly, standard curves were produced using six manufacturer provided standards (31.2–1000 pg/mL) and blanks by subtracting the optical density of the blanks from the standard solutions and plotting the standard solution optical densities. Gastrin concentrations for the samples were calculated using the equation generated by the standard curve for each corresponding plate. As this initial investigation aimed solely to determine the gastrin concentrations, sample duplicates were not performed. Assessment of parallelism, recovery, sensitivity, and specificity of this assay in this laboratory was not performed by the investigators. Gastrin was determined by this method within 18 months of the collection date (range 15–18 months) and within 30 d of RIA analysis.

### 2.4. ELISA Kit 2

Equine serum gastrin concentrations of seven samples (from six horses), as selected based on the gastrin range based on RIA results to represent low, medium, and high concentrations [22], were analyzed using a commercially available ELISA kit (Horse gastrin (GAST) ELISA kit, AFG Bioscience LLC, Northbrook, IL, USA). This kit utilizes a two-part sandwich ELISA method. The antibody or immunogen sources were not disclosed by the manufacturer. The manufacturer reported an assay detection range of 3–200 pg/mL with sensitivity of 0.5 pg/mL, intra-assay CV <8% and inter-assay CV <10% [23]. The sample number was determined to meet the industry recommendations of at least five duplicates per sample over at least three runs [22].

As for kit 1, analyses were performed by the primary investigator in a controlled laboratory, per the manufacturer’s instructions. Each of the seven samples was analyzed at least eight times per plate, over three plates. The standards were duplicated and standard curves were created for the average of the duplicated standards, which was used to determine the sample gastrin concentration. Per the manufacturer’s instructions, standard curves were produced using the provided standard diluted with the provided diluent to produce six standard solutions (15–270 pg/mL), using the sample diluent as a blank solution. The optical density of the standards was calculated by subtracting the optical density of the blanks, these were then averaged for the standard curve to be plotted. As for kit 1, sample gastrin concentrations were calculated using the equation generated by the standard curve for each corresponding plate. Assessment of parallelism, recovery, sensitivity, and specificity for this assay in this laboratory was not performed by the investigators. Gastrin was determined by this method up to 26 months after the collection date (range 23–26 months) using the stored samples as for ELISA kit 1. 

### 2.5. RIA

Equine serum gastrin concentrations were determined by RIA at a commercial laboratory, Canterbury Health Laboratories (Canterbury, Christchurch, New Zealand), using the previously validated RIA method [17]. The commercial laboratory reported inter-assay CV <11%, intra-assay CV <6%, recovery of 98–109%, and sensitivity of 3.3 pg/mL for this method. Commercial standards were adhered to, and internal validation was performed. However, these data are not available. Gastrin was determined by this method within 19 months of the collection date (range 16–19 months) from duplicate samples that remained stored at −20 °C until analysis. 

### 2.6. Data Analysis

Data analyses were performed using R version 4.3.1using RStudio [24] and the “Tidyverse” [25], “lmerTest” [26], “lme4” [26] packages and all plots were produced using “ggplot2” [27] and “ggpubr” [26]. The median and range for the results of each assay was calculated, and box plots of the results were produced. The Shapiro-Wilk test was used to determine the normality of the data. Log transformation was tested as a method of accounting for non-normality in the data. The correlation, between the RIA results and each of the ELISAs, was described using Spearman correlation and a linear model, with RIA as a response variable and the ELISA data as the predictor. For the kit 2 data, a similar linear model was built, with a horse-level effect included to assess its impact on the model. The assumption of normality of the residuals for all linear models was confirmed using quantile-quantile plots. A scatter plot of the correlation between kit 1 and RIA, as well as kit 2 and RIA, with a linear trend line, was also produced. The level of agreement between the RIA assay and each ELISA kit was analyzed using a modified Bland-Altman analysis [28,29,30,31] to account for the non-normality of the differences between the RIA assay and the two ELISA kits. An acceptable level of agreement between the two methods was determined prior to and included in the Bland-Altman plots for interpretation.

For each sample, the difference between gastrin concentration determined by RIA and both ELISA kits was calculated, as was mean concentration for each sample determined by RIA and each ELISA kit, respectively. The difference in concentration measured by RIA and each ELISA kit was then calculated as a percentage of the mean for each sample. These differences, both raw and as a percentage, were plotted against the means for each sample. Lines representing parity between the methods, indicating no bias, and 25% bias from the mean value, were also plotted to aid interpretation. A histogram of the differences was also produced and inspected to determine the distribution. Where strong bias was visible in the Bland-Altman plot, it was further characterized using a linear model.

## 3. Results

### 3.1. Summary

Equine gastrin concentration measured with the RIA (196 samples) had a median of 34.50 pg/mL (range 5–106 pg/mL) (Figure 1). The effects of omeprazole treatment are presented and discussed elsewhere [11]. The same samples, tested with ELISA kit 1 (196 samples), gave gastrin concentration measurements ranging from 2.01–756.67 pg/mL with a median of 186.25 pg/mL. Overall, the gastrin concentrations from ELISA kit 2 (7 samples) ranged from 4.07–36.66 pg/mL, with a median of 17.33 pg/mL (Figure 2). The range for each sample is shown in Table 1 alongside the respective RIA results for comparison. The distribution of the results from the assays RIA and kit 1 was determined to be non-normal (RIA: *p* < 0.01, Kit 1: *p* < 0.01). The results from kit 2 were normally distributed (*p* = 0.06). A retrospective analysis of power was performed using the results comparing each ELISA kit to the RIA, this showed greater than 80% power for both comparisons [32].

### 3.2. ELISA Kit 1

The coefficients of determination for the standard curves of the three plates were 0.987, 0.998 and 0.999 (3 s.f), respectively. The Spearman correlation between the gastrin concentrations measured with kit 1 and the RIA was 0.31 (Figure 3A). While the kit 1 results were a significant predictor of the RIA value (Table 2), the correlation was low and only 7% of the variation in the RIA results was explained by the kit 1 results, suggesting it is a poor-quality predictor. The mean bias (Kit 1-RIA) in kit 1 was 198.40 pg/mL (95% CI: −142.95–539.76). The positive bias in kit 1 increased as the mean concentration increased (Figure 3B,C). The differences between kit 1 and the RIA had a bimodal distribution (Figure 3D).

### 3.3. ELISA Kit 2

The coefficients of determination for the standard curves of the three plates were 0.988, 0.977 and 0.988 (3 s.f), respectively. The Spearman correlation between the gastrin concentrations measured with kit 2 and the RIA assay was −0.31 (Figure 4A). The horse-level effect was not significant (*p* = 0.60) and did not meaningfully affect the other coefficients, so it was excluded from the analysis in the interest of using the simpler model. There was a significant negative association between the kit 2 and RIA results; kit 2 explained only 10% of the variation in the RIA results (Table 2), indicating that kit 2 is a poor-quality predictor of RIA-measured gastrin concentration. The mean bias in kit 2 was −17.90 pg/mL (95% CI: −89.98–54.19), and the average bias for each sample is shown in Table 1. The negative bias in kit 2 increased as the mean concentration increased (Figure 4B,C). The differences between kit 2 and the RIA assay were split into three distributions, each of which was normally distributed (Figure 4D).

The Pearson correlation between the ELISA kit 1 and kit 2 was −0.299 (Figure 5).

## 4. Discussion

The current study retrospectively compared the quantification of equine serum gastrin concentrations between commercially available ELISA kits in a research laboratory, and RIA, performed in a commercial laboratory. The proposed hypothesis was not supported, with a poor correlation of ELISA measurements using both kits compared to RIA. The negligible positive correlation of ELISA kit 1 and the low negative correlation of ELISA kit 2 to the RIA measurements renders them impractical for the measurement of equine serum gastrin [33]. Furthermore, the bias of both ELISAs is greater with increasing gastrin concentration, making the use of these assays for the assessment of hypergastrinemia inappropriate.

Despite the poor correlation between ELISA and RIA, the coefficient of determination for the standard curve of the two ELISA kits was excellent in the present study, ranging from 0.977–0.999 (3 s.f). This suggests that the ELISA kits accurately measured the provided matrix-free control solutions, but not the gastrin in the serum samples. This might have contributed to the variation seen, as the effect of matrix on antibody-based assays can be a significant issue [34]. It is widely recommended that blanks and standards used for the production of calibration curves be composed of the same matrix as the samples tested (e.g., species-specific hormone-free serum), to accommodate for the variability introduced by the matrix [22].

Another source of potential variability is species-specific variations in the composition of gastrin. The amino acid composition of equine gastrin, namely G-34 and G-17, varies from the human forms by three and two amino acids, respectively [35]. Importantly, both equine gastrin peptides lack the highly acidic chain of amino acids preceding the bioactive terminal of other species, including man, making equine gastrin less acidic [35]. In contrast to other species, equine gastrin has been found only in the unsulfated form [35]. These inherent differences between equine gastrin and gastrin from other species might affect stability and analysis in unknown ways. An investigation into the accuracy of commercial kits for the measurement of human gastrin found seven of twelve kits to be inaccurate for this purpose, with errors including failure to bind more than one of the bioactive forms of gastrin, and increased reaction to unknown matrix compounds [36]. Whether such an effect was present in the current study is not known.

Another potential source of variation is operator-related variability. In an attempt to limit errors in ELISA analysis, variables kept consistent included the use of a primary investigator (J.R.V.) performing all ELISAs under the supervision of another investigator (K.R.G.), all laboratory equipment used was consistent, and the laboratory was strictly climatically controlled. The high coefficients of determination consistently seen for both ELISA kits suggest that operator-related variability was minimal. Data regarding the variability in assay technique by the technician performing the RIA is not available from the commercial laboratory. 

A limitation of the present study is the possible effect of the sample storage time on results. All samples were stored under the same conditions for up to 26 months from collection. The RIA and initial ELISA kit analysis were carried out within a month, whereas the second ELISA kit was analyzed up to 8 months later. Samples were stored in duplicate, with different sets used for ELISA kit 1 and RIA. These were stored, and handled, under identical conditions to limit any possible effect of prolonged storage. For analysis using the ELISA kit 2, the samples used were the same as for the ELISA kit 1 so an effect due to prolonged storage compared to RIA cannot be excluded. In performing duplicate measurements with the ELISA kit 2, the intra- and inter-assay variation cannot be attributed to possible storage effects. Human gastrin is stable in serum at −20 °C for up to 5 years [20], but this has not been studied for equine gastrin. Both ELISA kits recommended processing samples within 6 months of collection, however the foundations of such a recommendation are not clear. The single time-points of sample analysis for each method in this study preclude any assessment of the stability of equine gastrin in stored serum samples over time. This is a limitation of the present study and further research into the stability of equine gastrin is warranted.

Sample handling can also affect results. The time from blood sampling to processing of the serum has a significant influence on the stability of the human gastrin, and processing for storage no more than 6 h after sample collection is recommended [37]. The current study had all samples processed and stored within 4 (range 1–4) hours of collection. Additionally, the time from collection to storage was consistent for each sample, regardless of the assay. It is possible that, in the time from thawing to measurement, degradation of gastrin or other alterations of the matrix might have influenced the assay measurements. However, in this study the time from thawing to assay was limited as much as possible. Furthermore, this possible effect would be expected to alter all samples within a plate similarly which was not reflected in the results and any delay present reflects the real-world usage of such kits.

The comparison of two ELISA kits to the RIA should also be considered as a limitation of the current study. Although the RIA method has been previously validated with equine serum, and the commercial laboratory is expected to follow stringent guidelines, the authors cannot guarantee that the technique used by the commercial laboratory was exactly as previously described [17]. Furthermore, the study design itself is considered a limitation of the present study. Due to the initial study design not quantifying duplicated samples with ELISA kit 1, it is possible that mean gastrin measurements over multiple duplicates (as performed for the ELISA kit 2) might have correlated to the RIA better. Future research should aim to thoroughly validate these assays for quantification of equine gastrin in serum in accordance with industry guidelines.

This study compared two different, commercially available, non-validated gastrin ELISA assays to the previously validated RIA for equine serum. The kits used represent the only commercial ELISAs that these authors are aware of. Both kits were used with strict adherence to the manufacturers’ recommendations, except for the sample storage time as discussed previously. Wide recognition of the difficulties of antibody-based research in the last decade, termed the ‘reproducibility crisis’, have raised concerns in human research leading to stricter recommendations regarding the use of antibody-based techniques in research [36]. To demonstrate the long-term reproducibility of sound antibody-based assays, the revalidation of the human gastrin RIA has been reported, with similar sensitivity and improved sensitivity compared to the original assay more than 40 years prior [34]. It is recommended that research laboratories raise their own antibodies or acquire custom-generated antibodies, rather than using small aliquots or small batches of ELISA kits [34]. In veterinary research, the practice of raising specific antibodies for use in assays is less commonly accessible, therefore, these authors propose the prioritization of selecting validated methods in quantifying equine gastrin. The authors encourage future researchers to consider the limitations of both the ELISA and RIA techniques when using them to quantify equine gastrin, as maximizing the results of these assays is likely to benefit future equine gastric research. Until validation can be performed, and further research into hypergastrinemia in horses is undertaken, these techniques are unlikely to be of clinical use. 

## 5. Conclusions

With poor agreement between ELISA and RIA, when the interval from sample collection to analysis is delayed, this study fails to demonstrate ELISA as a robust method to quantify serum gastrin in the horse. Future studies should attempt to validate an alternative equine gastrin assay to RIA, with a study design paying close attention to the factors influencing variability and the effects of mitigating actions. Until an alternative method of serum gastrin quantification in the horse can be validated, unvalidated techniques such as the ELISA should not be adopted over the existing reference standard of RIA. 

## Figures and Tables

**Figure 1 animals-14-02937-f001:**
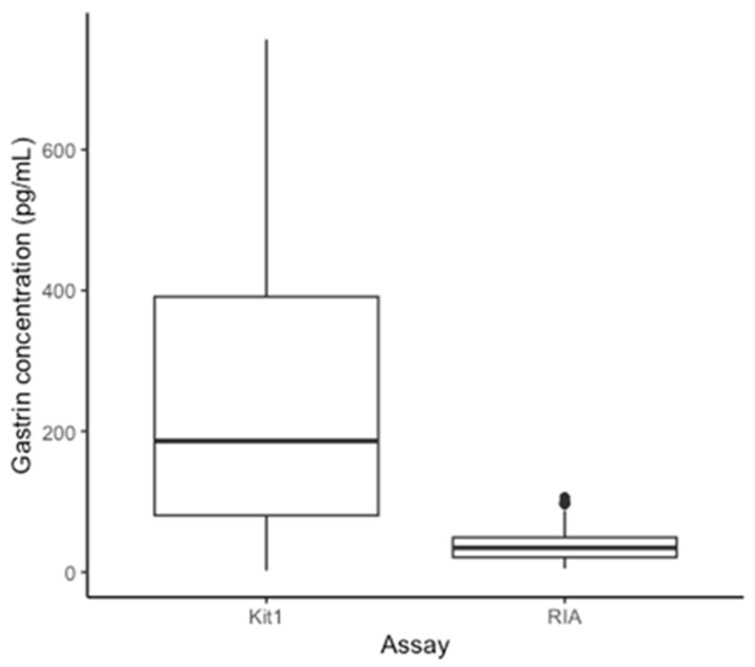
Box plot to demonstrate the median, interquartile range, and range of gastrin concentrations determined by RIA and ELISA kit 1 for all 196 samples.

**Figure 2 animals-14-02937-f002:**
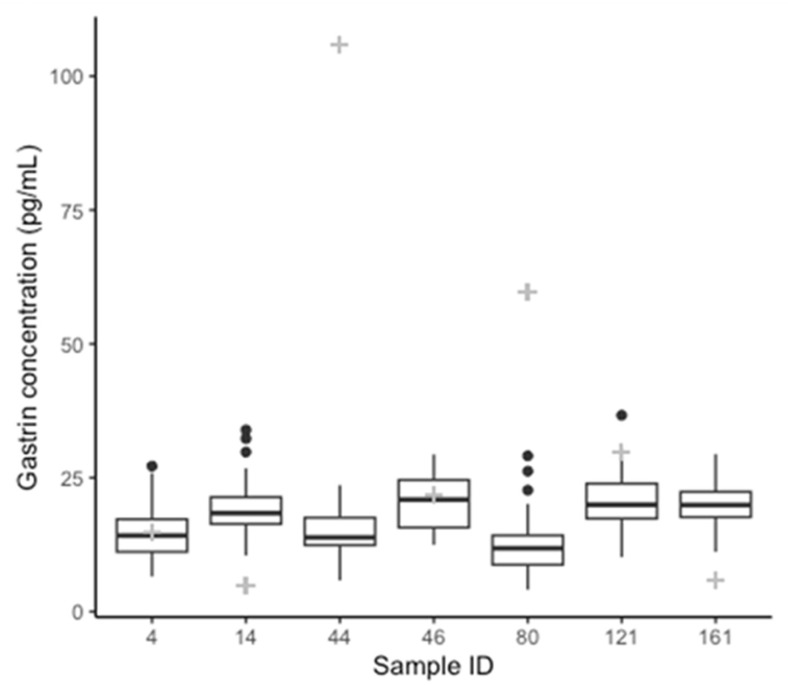
Box plots showing median, interquartile range, and range of gastrin concentrations for each of the seven samples tested in replicate by ELISA kit 2. Gastrin concentration determined by RIA for each sample is indicated by the grey “+” for comparison.

**Figure 3 animals-14-02937-f003:**
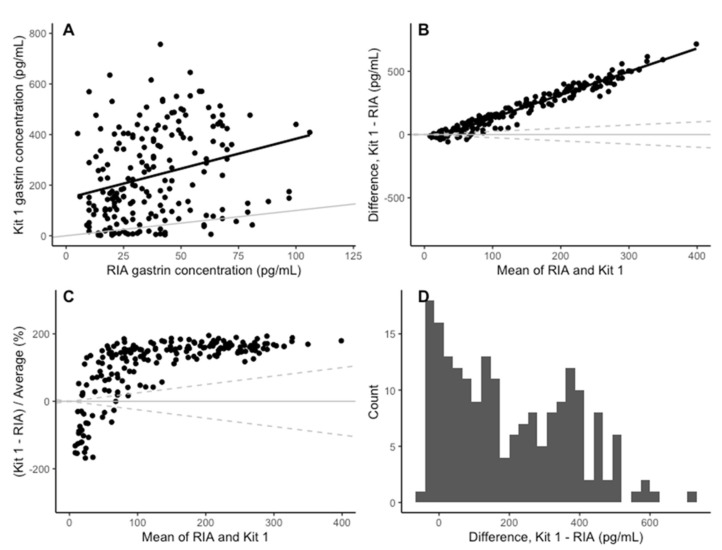
(**A**) Comparison of the concentration of gastrin detected in the RIA and ELISA kit 1. The grey line indicates parity and the black line indicates line of best fit. (**B**) The change in the difference between RIA and ELISA kit 1 as the mean concentration increases. The solid black line indicates the mean difference between RIA and ELISA kit 1. The solid grey line is y = 0 and the dashed grey lines represent a ±25% bias from the mean of the two assays. (**C**) The change in difference in gastrin concentration detected by RIA and ELISA kit 1, as a percentage on the mean concentration, as the mean concentration increases. The solid grey line indicates y = 0 and the dashed grey lines indicate a 25% level of bias. (**D**) Histogram of the difference between RIA and ELIAS kit 1 for each sample.

**Figure 4 animals-14-02937-f004:**
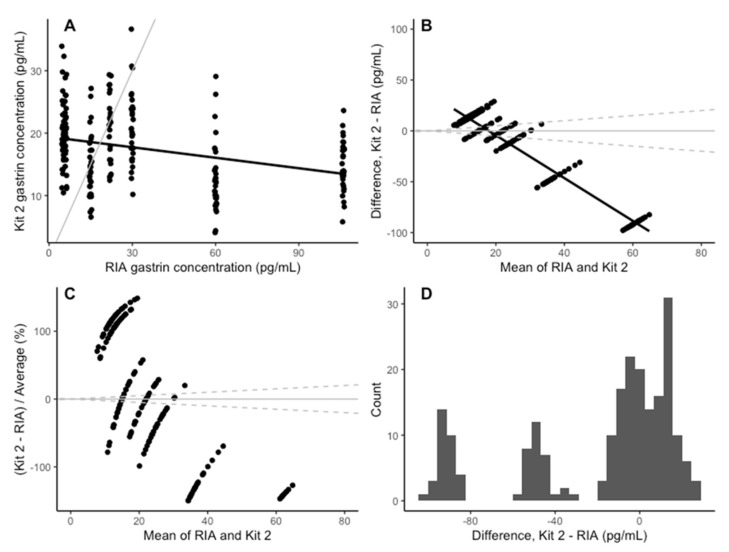
(**A**) Comparison of the concentration of gastrin detected in the RIA assay and the ELISA kit 2. The grey line indicates parity, and the black line indicates line of best fit. (**B**) The change in the difference between RIA and ELISA kit 2 as the mean concentration increases. The solid black line indicates the mean difference between the RIA and the ELISA kit 2. The solid grey line is y = 0 and the dashed grey lines represents a ±25% bias from the mean of the two assays. (**C**) The change in difference in gastrin concentration detected by the RIA and the ELISA kit 2, as a percentage on the mean concentration, as the mean concentration increases. The solid grey line indicates y = 0 and the dashed grey lines indicate a 25% level of bias. (**D**) Histogram of the difference between the RIA and kit 2 for each sample.

**Figure 5 animals-14-02937-f005:**
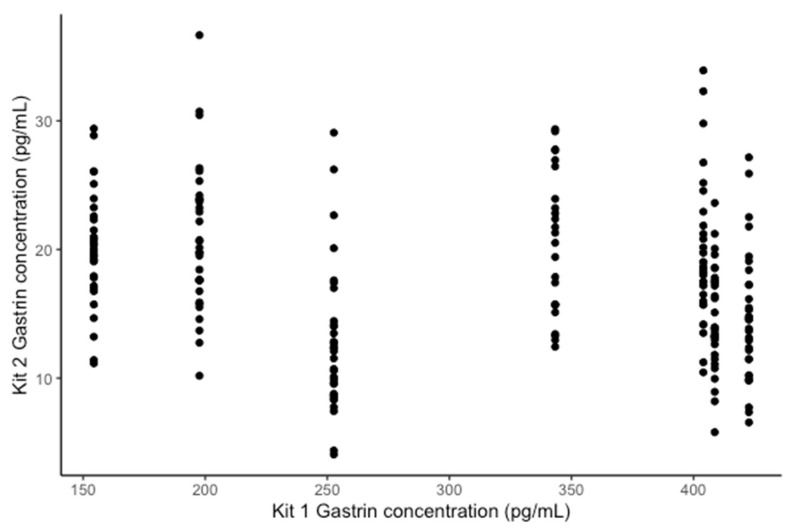
Comparison of the concentration of gastrin detected by the ELISA kit 1 and kit 2.

**Table 1 animals-14-02937-t001:** Gastrin concentrations determined by RIA and ELISA kit 1, and for each of the seven samples tested in replicate with ELISA kit 2, showing the difference between the three methods. The number of replicates for each of the seven samples measured with ELISA kit 2 is indicated in the final column.

Sample	RIA Gastrin (pg/mL)	Kit 1 Gastrin (pg/mL)	Mean Kit 2 Gastrin (95% CI) (pg/mL)	Kit 2 Gastrin Range (pg/mL)	RIA-Kit 2 (95% CI) (pg/mL)	Number of Kit 2 Replicates
14	5	404	19.55 (8.81–30.29)	10.45–33.92	14.55 (3.81–25.29)	32
161	6	154.38	20.05 (11.47–28.63)	11.15–29.4	14.05 (5.47–22.63)	32
4	15	422.67	14.59 (4.74–24.44)	6.56–27.16	−0.41 (−10.26–9.44)	32
46	22	343.33	20.52 (9.67–31.37)	12.44–29.36	−1.48 (−12.33–9.37)	24
121	30	197.62	20.76 (9.72–31.8)	10.18–36.66	−9.24 (−20.28–1.8)	32
80	60	252.67	12.72 (1.57–23.87)	4.07–29.08	−47.28 (−58.43–36.13)	32
44	106	408.67	14.64 (6.82–22.46)	5.8–23.62	−91.36 (−99.18–83.54)	32

**Table 2 animals-14-02937-t002:** Parameters of linear models used to describe the correlation between the gastrin concentrations determined by RIA and kit 1, RIA and kit 2, and to describe the change in bias in the difference in measured gastrin concentration changes with the mean gastrin concentration.

Model	Parameter	Coefficient Estimate	Std. Error	*p* Value	R Squared
RIA-Kit 1	Intercept	30.19	2.36	<0.01	0.07
	Kit 1	0.03	0.01	<0.01	
RIA-Kit 2	Intercept	67.8	6.88	<0.01	0.1
	Kit 2	−1.86	0.37	<0.01	
RIA & Kit 1 mean-RIA & Kit 1 difference	Intercept	33.83	2.21	<0.01	0.95
	RIA & Kit 1 difference	0.52	0.01	<0.01	
RIA & Kit 2 mean-RIA & Kit 2 difference	Intercept	18.76	0.4	<0.01	0.9
	RIA & Kit 2 difference	−0.42	0.01	<0.01	

## Data Availability

The original contributions presented in the study are included in the article, further inquiries can be directed to the corresponding author.

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
