# Peer review of "Evaluation of Two Commercial ELISA Kits for Measuring Equine Serum Gastrin Compared to Radioimmunoassay"

_animals, 2024, doi:10.3390/ani14202937_

Round 1

Reviewer 1 Report

Comments and Suggestions for Authors

In this peer-reviewed paper, the authors evaluate the utility of two commercially available ELISA tests (for measuring gastrin levels) and contrast it with data obtained by radioimmunoassay analysis. They made their assessment based on samples taken from horses.

From a substantive point of view, this is not a research paper. I am unable to understand the point of this work and what novelty it brings to the current state of knowledge. The advantages and disadvantages of both ELISA and RIA have been known for years. The hypothesis that the two tests are comparable in quality is simply trivial. There is not a single piece of science in this publication. For this reason, I cannot recommend the paper for publication in a scientific journal of the kind Animals is (the paper does not fit the scope). Yes the work is suitable for publication, but in a journal of a methodological nature.

Also, I don't understand the point of blacking out important information. Without key data on the product, ethics committee approval, etc., there is no serious evaluation of the work.

Author Response

Thank you for your comments. Given the continued use of non-validated gastrin ELISAs in equine research as shown in the introduction, the authors believe that awareness of the concerns raised in the present work would be of benefit to future researchers in this field. As the design of the current study was driven in part by initial findings and compounded by processing delays, the authors recommend more stringent evaluation of the ELISAs in the future and believe that this work does not constitute a validation study and hence would not be suitable for a methodical journal. Rewording to make this clearer has been performed.

The authors regret that the version of the manuscript shared to reviewers contained blacked out information and have rectified this in the resubmission. We hope that this, along with the changes made, will allow you to have a deeper understanding of the work performed.

Reviewer 2 Report

Comments and Suggestions for Authors

Please address the feedback and issues highlighted in the annotated manuscript.

Author Response

Thank you for your comments, we have used them to make improvements throughout. Please see point-by-point responses below. 

Comment 1: What is meant by non-validated? These two ELISA were created for use with horse serum, so surely these immunoassays were fit for purpose.

- The ELISA kits are reported by manufacturers to be internally validated; however, this information is not available to consumers. We have elaborated on line 61 to make this clearer to the reader.

Comment 2: Blacked out sections but can see the authors so need to fix.

- All redacted sections have been removed.

Comment 3: Although both these ELISA kits were commercially available for use in horses and it can be assumed that appropriate quality controls were used to validate the kits before commercialization – were recovery, parallelism, intra and inter assay variation etc carried out by the authors to establish, for themselves, the reliability of these commercial kits within this laboratory.

- These were not, statements to this effect have been added to the respective methods sections and these study design limitations further emphasised in the discussion.

Comment 4: There needs to be much more detail here regarding this RIA, there is no indication here that the RIA is suitable for use, other than the paper published in 1988. The authors seem to indicate that this RIA is the ‘gold standard’ against with the ELISA kits should be compared, yet there is no indication the RIA is used routinely to measure equine gastrin. The same information regarding assay detection range, sensitivity and intra and inter assay need to be included as has been for the two…

- Thank you for raising this important point, we have contacted the commercial laboratory for more information and have provided this in the manuscript (lines 144-145). No further information is available which has been discussed further, and comparison to RIA has been explicitly stated as a limitation of the present study.

Comment 5: Why not compare both the ELISA kits for correlation? If the two kits is strong – perhaps the RIA is not ideal. As there is no indication of what antibody was used in the RIA or any other details, why should readers be confident that the RIA is giving the ‘correct’ answer?

- We have added the correlation between the two kits to the results. Unfortunately, these details for the RIA are not available upon contacting the commercial laboratory.

Comment 6: It is unclear why this degree of replication was not also completed for the RIA and Kit 1. Could this be clarified?

- Due to the initial aim of the study to quantify gastrin using the ELISA kit with comparison to RIA as quality control, the design did not include duplicates for the first kit. After these results were analysed, the methods were adjusted before applying to Kit 2. This is a limitation of the present study, lines 112-113 describe this and this has been more explicitly stated in the discussion as highlighted.

Comment 7: Were the quality control samples run by this operator to show the intra and inter assay variation for all immunoassays? This is the true test for how well the technician can conduct both the ELISA and RIA. Please include intra and interassay variations from the laboratory, not just the manufacturer’s information.

- As the previous work done by this technician in this research laboratory has been using unvalidated methodology the inter- and intra-assay variability from previous work is likely not representative of technique. Because of this we have presented the coefficients of determination for these plates as they represent both the technician’s ability and the kits used which is more relevant for the study. These coefficients of determination presented in the results and discussed later are for ELISAs carried out by the investigator in the present study. Unfortunately, data for the technician carrying out the RIAs at the commercial laboratory is not available due to their privacy policies.

Reviewer 3 Report

Comments and Suggestions for Authors

In general, well described study design, only some minor questions raised up.

Line 24: selected samples ? random or specific selected samples?

Line 57-58: is this technique the ‘golden standard’? Is there information available about validation of both Elisa Kits in horses? Were manufacturers contacted to have information about validation of the kits in horses?

Line 89: what is the stability of a sample for gastrin measurement? Add information in the introduction about degradation during storage or in this part.

Line 160: of all samples – over the complete study? Describe the impact of omeprazole treatment?

Line 164 : of 7 samples

Line 160-164: please add here total number of samples for each test between brackets – will be easier to follow

Author Response

Thank you for your comments, we have used them to make improvements throughout. Please see point-by-point responses below.

Line 24: selected samples ? random or specific selected samples?

  • Specifically selected to represent a range of measurements. This is defined on lines 106-107 but has been left out of the abstract to meet the word count

Line 57-58: is this technique the ‘golden standard’? Is there information available about validation of both Elisa Kits in horses? Were manufacturers contacted to have information about validation of the kits in horses?

  • The ELISA kits do not have publicly available validation data for horses and the manufacturers did not disclose information beyond that which is included in the text when contacted by the authors. The RIA is widely considered the reference standard as it is the only technique to quantify equine gastrin that has published validation; however, the RIA is still not a 100% accurate test. The comparison to RIA as a limitation of the study has been added in the discussion.

Line 89: what is the stability of a sample for gastrin measurement? Add information in the introduction about degradation during storage or in this part.

  • Added as requested (now lines 93-94).

Line 160: of all samples – over the complete study? Describe the impact of omeprazole treatment?

  • The effects of omeprazole are discussed in another publication, this has been clarified and referenced to adhere to word limits.

Line 164 : of 7 samples

  • Changed as requested.

Line 160-164: please add here total number of samples for each test between brackets – will be easier to follow

  • Changed as requested.

Reviewer 4 Report

Comments and Suggestions for Authors

This paper is about the evaluation of two commercial ELISA kits for measuring equine serum gastrin compared to radioimmunoassay, with a focus on the importance of gastrin in gastric disease in horses and the limitations of non-validated ELISA assays.

The introduction section effectively provides background information on the significance of gastric mucosal diseases in horses and the role of gastrin in gastric acid secretion with a focus on rebound gastric hyperacidity. The authors appropriately highlight the need for reliable assays to measure equine serum gastrin concentrations, setting the stage for their comparison of ELISA kits against a validated RIA method. The hypothesis is clearly articulated.

The materials and methods  provide detailed account of the experimental design, animal selection, sample collection, and analytical procedures used in the study. The description of the ELISA kits, RIA method, and statistical analysis is thorough and allows for reproducibility of the experiment. However, the power calculation, which is crucial for determining the adequacy of the sample size, is inadequately described. The authors merely state that the number of horses selected was based on a power calculation described elsewhere without providing the necessary details and the reference seems to be unnecessary.  Please include a concise summary of the power calculation within the methods section to ensure transparency.

The results  present the findings in a well structured manner, the data presentation is clear, and the use of figure makes results easy to understand.

The discussion section critically analyzes the study’s findings, acknowledging the poor correlation between the ELISA kits and the RIA method, and the variability in results with increasing gastrin concentrations. The authors do well to explore potential sources of variability, such as species-specific differences in gastrin composition and operator-related factors. However, the discussion lacks a robust clear critique of the implications of these findings for the broader field of equine gastric research. Please clearly articulate the limitations of your study. You describe them without actually mentioning they are limitations, such as the potential impact of sample storage on assay performance. While these issues are mentioned, they are not explicitly labeled as limitations, which diminishes the clarity and impact of the discussion. The discussion would benefit from a more explicit consideration of how these findings affect the use of ELISA kits in future research and clinical practice

The paper provides valuable insights into the performance of commercially available ELISA kits for measuring equine serum gastrin concentrations compared to a validated RIA method. Overall I enjoyed reading presentation of negative results/ a refuted theory as they are valuable information and very often neglected.  

Author Response

Thank you for your comments, we have used them to make improvements throughout. Please see point-by-point responses below. 

...However, the power calculation, which is crucial for determining the adequacy of the sample size, is inadequately described. The authors merely state that the number of horses selected was based on a power calculation described elsewhere without providing the necessary details and the reference seems to be unnecessary.  Please include a concise summary of the power calculation within the methods section to ensure transparency.

  • Power calculation for the outcomes were not performed prospectively; however, we have done a retrospective analysis of power detailed on lines 188-189 to asses this. We have also clarified in the materials and methods that the power calculation performed prospectively was for another outcome to make this clearer to the reader.

However, the discussion lacks a robust clear critique of the implications of these findings for the broader field of equine gastric research. Please clearly articulate the limitations of your study. You describe them without actually mentioning they are limitations, such as the potential impact of sample storage on assay performance. While these issues are mentioned, they are not explicitly labeled as limitations, which diminishes the clarity and impact of the discussion. The discussion would benefit from a more explicit consideration of how these findings affect the use of ELISA kits in future research and clinical practice.

  • Thank you for raising this important point. The discussion has been reworked to address the implications of the concerns raised previously and to further highlight the limitations specifically.

Reviewer 5 Report

Comments and Suggestions for Authors

Generally well written and presented. Even though there are no clear insights into why the ELISA kits have such poor performance, it is important that these results are published.

The methodology and presentation of results is well executed.

There are a few minor issues to be addressed:

71: I am assuming that the redacted sections will be included in the published paper? 

75: although the number of horses used was calculated for the concurrent study, in light of the negative findings for the ELISA tests there should be some analysis or statement to indicate that the sampling for this study had sufficient power to detect significance, especially for ELISA kit 2.

100 & 117: Indicate how the standard curves were produced in regards to the substrate used.

103 & 119: Indicate why there was such a long period after collection.

156-158: Is this an editing comment that was left in?

279: indicate the range of times from thawing to analysis.

Comments on the Quality of English Language

No major issues with the quality of English, except some minor corrections to be done.

Author Response

Thank you for your comments, we have used them to make improvements throughout. Please see point-by-point responses below. 

71: I am assuming that the redacted sections will be included in the published paper? 

  • The redacted sections have been removed.

75: although the number of horses used was calculated for the concurrent study, in light of the negative findings for the ELISA tests there should be some analysis or statement to indicate that the sampling for this study had sufficient power to detect significance, especially for ELISA kit 2.

  • We have performed a retrospective analysis of power which we have included in lines 188-189.

100 & 117: Indicate how the standard curves were produced in regards to the substrate used.

  • Standard curve determination has been detailed for both ELISA kits (lines 107-111 and 131-136).

103 & 119: Indicate why there was such a long period after collection.

  • This study was started late in the COVID-19 pandemic with many restrictions still in place in the authors’ countries of residence. This has been described in lines 90-92 for the readers.

156-158: Is this an editing comment that was left in?

  • Yes, and has been corrected.

279: indicate the range of times from thawing to analysis.

  • Added as requested.

Round 2

Reviewer 1 Report

Comments and Suggestions for Authors

Unfortunately, the authors did not reliably address my comments. The response is laconic and does not explain much.  Actually, everything has already been said in my previous review. This article is not a research article (which the authors actually agree with) and is of little scientific value. It has not undergone a substantial revision, and therefore it should be rejected.

Author Response

We are uncertain how we can better address your comments. 
As we identified, the routine use of non-validated gastrin assays means that a manuscript such as this highlighting some of the variation associated with ELISA assays is pertinent. We appreciate we opportunistically performed the comparisons and have highlighted this as a limitation with the study, and have not over interpreted the relevant data. The intention with the manuscript is to highlight the variation and provide a signpost within the literature that the direct comparisons of values between ELISA assays should be performed with caution. We appreciate that this is not ground break science – but feel it is important to have this signposted within the literature given the variety of assays used in different studies.

Reviewer 2 Report

Comments and Suggestions for Authors

Thank you to the authors for diligently addressing my initial concerns. I appreciate dealing with a commerical laboratory does have its limitations. It is refreshing to have researchers publishing negative results to alert others who may inadvertently follow the wrong path.

Please accept my condolences for the loss of your colleague.

Author Response

Thank you for your time and comments.